# Implementing the Concept of Neurolinguistic Programming Related to Sustainable Human Capital Development

**Miroslav Frankovský [1], Zuzana Birknerová [2], Róbert Štefko [3] and Eva Benková [4,***

[1] Department of Managerial Psychology, Faculty of Management, University of Prešov in Prešov, Konštantínova 16, 080 01 Prešov, Slovakia

[2] Trade Department of Managerial Psychology, Faculty of Management, University of Prešov in Prešov, Konštantínova 16, 080 01 Prešov, Slovakia

[3] Department of Marketing and International Faculty of Management, University of Prešov in Prešov, Konštantínova 16, 080 01 Prešov, Slovakia

[4] Department of Intercultural Communication, Faculty of Management, University of Prešov in Prešov, Konštantínova 16, 080 01 Prešov, Slovakia

* Correspondence: eva.benkova@unipo.sk

**Abstract:** Sustainability business is a multidimensional construct the research of which assumes an interdisciplinary approach. The given approach is related to a holistic concept of defining sustainability business. From the point of view of this concept, it is necessary to understand economic, socio-cultural attributes, and the human resources attributes as one whole that differs from the final study results of individual elements of sustainability business in the meaning of the Principle of Emergence. Within this concept, it is possible and inevitable to focus our attention on the issue of human capital in the context of socio-cultural factors. In the presented paper, attention was paid to the results of examining the neurolinguistic programming (NLP) as one of the possibilities of sustainable development of human capital. The main aim of the research with 104 respondents was to identify and specify the indicators of sustainable human capital development and selected attributes of neurolinguistic programming. The results of analyses confirmed the existence of these connections. The respondents who evaluated the NLP attributes more positively also evaluated more positively the indicators of a sustainable human capital development. The limits of the presented analyses results are limited by the cultural conditions of the research, size, and the selection of the research sample and also by accepting other factors in the context of a holistic approach to this issue.

**Keywords:** neurolinguistic programming (NLP); sustainable development; human capital

## 1. Introduction

Defining the issue of sustainable human capital development must be understood as part of the sustainability business in terms of focusing attention primarily on the acquisition of special knowledge, abilities, skills, experience and this through education [1]. This concept must be defined more comprehensively in terms of a holistic approach to the issue of sustainability business. According to Mazouch and Fisher [2], the level of human capital is influenced by a number of interacting factors. Attention is paid not only to training and practice, but also to health, physical and mental condition, nutrition, lifestyle, changes in thinking, etc. [3,4].

Defining sustainable human capital is associated with a competitive advantage and economic growth, especially in times of great turbulences and constant changes [5]. Investing in the sustainable development of human capital is linked to the efficient use of resources found in each of us and to the

expectations of future benefits [1]. These investments ultimately increase the quality of life for both, individuals and a society [6].

The sustainable development of human capital is represented by a rich mosaic of possibilities and approaches. One common denominator is focusing the attention to a man as an individual being in the sense of humanistic and existential concepts. From this point of view, it is necessary to draw the attention to the issue of flexibility of people's thinking with accepting the necessity of changes in thinking. Identifying and specifying the attributes of neurolinguistic programming is one way to look at the issue of sustainable human capital development from a more comprehensive perspective, with an emphasis on people's thinking.

## 2. Literature Review and Hypotheses

The concept of human capital sustainability with the aim of sustainable economic development can be defined as sustainable management of high economic, environmental, and socio-cultural standards for present and future generations [7]. In modern science, economic theory and practice, the term sustainable development has a global origin and a complex multilateral character. It takes into account different opinions of authors and market factors by which the tendency of global competitiveness, technological and scientific progress is growing [8] which are directly related to the sustainable development of human capital. Sustainable development is a continuous process [9]. The concept of sustainability is constantly formulated by many aspects that affect it. As society evolves, the concept of sustainable development must accept new challenges that need to be addressed [10]. Even in terms of sustainable human capital development, a generational perspective is emerging to the forefront [11].

The concept of sustainable human capital development requires sustained investment effort and high efficiency. As an innovative element, it is a key to a sustainable economic development and growth [12]. The primary structural component of sustainable development in the economic dimension is the global and unified connection between individual parts of the sustainability business subsystems. National economies, international economic relations, the global division of labor, or the world economy have the ability to create a new concept of sustainable development that will also cause structural changes in the social area with the impacts and demands on the level of human capital [13]. Furthermore, from this point of view, the role of a social system is becoming important, aligning the economic, environmental, social, and institutional component of sustainability considering the specifics of a given society [14]. Current economic development should favorably influence the next economic development for future generations [15].

In the concept of sustainable human capital development, neurolinguistic programming can be seen as an important factor related to the flexibility of people's thinking. Neurolinguistic programming is a concept, the essence of which is the analysis of the individual perception of the world in order to increase the success of people. This concept can be characterized as a paradigm of relationships between communication, a way of thinking through different techniques for improving communication, and changing behavior towards achieving a goal [16]. It reflects a holistic view of a body and mind, connection of individual neurological processes to a language, and learned behavioral strategies [17].

Neurolinguistic programming is a process of modeling unique, each person's own conscious and unconscious patterns of thinking, behavior, and communication, while the individual continuously develops and strengthens his or her own potential [18]. When setting personal goals, it is appropriate to consider the socio-economic and cultural environment which the individual lives in. Our perception filters, beliefs, values, language, and culture are developed through our experience of the world [19].

Neurolinguistic programming (NLP) deals with goal-setting techniques with regard to their definition. The more specific and positive the goals, the more it is possible to program the brain in terms of finding opportunities [20]. According to Tripathi [21], individuals using NLP can understand emotional and behavioral patterns of behavior. Through NLP, they can achieve results they thought were unattainable or overcome obstacles in their personal growth.

Neurolinguistic programming is dedicated to how to achieve unique results and how to excel [18], gradually finding more and more ways and tools to identify and recognize the factors allowing being excellent. The author states that NLP is one of the most effective methods in human development and learning. NLP has a broad-spectrum use [22], from the point of view of individuals, organizations, society as a whole, as well as in the context of sustainable human capital development.

Based on the theoretical knowledge as a result from our research [23], the objective of the presented research project was determined. The aim of the research is to identify and specify the indicators of sustainable human capital development as well as the connections between the extracted factors of sustainable human capital development and selected attributes of neurolinguistic programming-communication (NLP-C) and neurolinguistic programming-techniques (NLP-T). Based on the goal, we set out the following research hypotheses:

**H1.** *We assume that it is possible to identify and specify the indicators of the sustainable development of human capital;*

**H2.** *We assume that there are statistically significant relationships between assessing selected NLP-C attributes and the indicators of sustainable human capital development;*

**H3.** *We assume that there are statistically significant relationships between assessing selected NLP-T attributes and the indicators of sustainable human capital development.*

## 3. Materials and Methods

We verified the hypotheses using three methodologies: NLP-C, NLP-T, the indicators of sustainable human capital development—SHC.

**NLP-C: Neurolinguistic programming-communication** [23]:

It identifies communication skills of individuals and the importance of communication. The methodology contains 17 items that are assessed on a 5-point Likert scale. Four factors were extracted by the factor analysis:

Asking questions—one of the important communication skills is the art of asking the right questions in an appropriate way, while obtaining, specifying, and verifying information but also supporting or blocking individual communication.

Active listening—it enables to verify the correctness of interpreting a message by a communication partner. Conversely, the reluctance to listen, suppression of listening by own narration, interrupting the speech, verbal and non-verbal expression of interference diminish the effectiveness of communication or they disturb it.

Body language—identification and interpretation of non-verbal communication expresses the emotions and inner attitudes of a man, sometimes more than spoken speech. It is the so-called first impression which can reveal a lot to an experienced observer.

Assertiveness—it is based on natural human behavior and we can adapt it to ourselves and our needs. The goal of assertive communication in the sense of healthy self-esteem is the authentic expression of emotions and adequate communication of one's own attitudes and demands and to remain in one's own places.

**NLP-T: Neurolinguistic programming-techniques** [23]:

It identifies the techniques f neurolinguistic programming. The methodology contains 15 items that are assessed on a 5-point Likert scale. Three factors were defined by the factor analysis:

Representational systems—believing that every human being prefers one of the five sensory systems. Every person has his/her own, the so-called preferred sensory system, which one uses the most and in which one feels the best.

Rapport—it creates an ideal state of communication based on trust and understanding that managers can use when influencing behavior, initiating changes, and persuading people.

Communication is one of the principles of success and it can be defined as creating a spirit of trust and respect among people to create a greater probability of co-operation.

Leading—leadership, guidance, or management: Adapting and creating a harmony with a partner for a certain period of time should lead to a sufficient relationship that will then enable to lead a partner to wherever we want. In this way, we can use partner's different representational systems.

**SHC—indicators of sustainable human capital development:**

The basic assumptions in NLP are associated with the development of human capital that needs to be assessed in terms of sustainable development. In this context, we present the original SHC methodology that was verified in the research. The methodology consists of 11 statements related to the sustainable development of human capital on which NLP relies. Each statement contains 6 possible ways of responding to the degree of acceptance of it (0-definitely no, 1-no, 2-rather no than yes, 3-rather yes than no, 4-yes, 5-definitely yes). As an example, we present the following statement: "There is no failure, there is only a feedback", "I have all the resources I need to make a change".

Using a factor analysis and Principal Component method with Varimax rotation, we have extracted 2 factors: Uniqueness, positivity (Hypothesis 1).

In the research presented, the data were collected from 104 respondents, out of whom 48 were women (46.2%) and 56 men (53.8%), 80 managers (76.9%) and 24 executives (23.1%). The average age of the respondents was 40.20 years (standard deviation was 10.187 years) ranging from 22 to 63 years. The number of years worked by the respondents was on average 17.83 years (the standard deviation was 10.564 years), ranging from 1 year to 40 years. All respondents completed the NLP course.

## 4. Results

The date obtained enabled to perform the factor and correlation analysis using the SPSS 22 statistical program, related to the studied issue that are described in the structure of the formulated hypotheses.

Identification and specification of the indicators of sustainable human capital development:

The aim of the research was to verify the SHC methodology through hypothesis 1, allowing identifying and specifying the indicators of sustainable human capital development based on a psychometric approach. Using a factor analysis based on KMO test (0.812) and Bartlett's test (586.3, sig. 0.000), the Principal Component method with Varimax rotation was used. Two factors were extracted (Figure 1, Table 1):

1. Uniqueness: It is assessed from the point of view of effective communication even without words. Unique individuals have their body and mind in harmony and perceive the world in a unique way. They are not afraid of changes since they believe that they dispose of all the needed resources;
2. Positivity: Is perceived in terms of success, positive direction, and intention. Positive individuals work best as they know in a given situation. They always have a choice to change their lives when other individuals have done it.

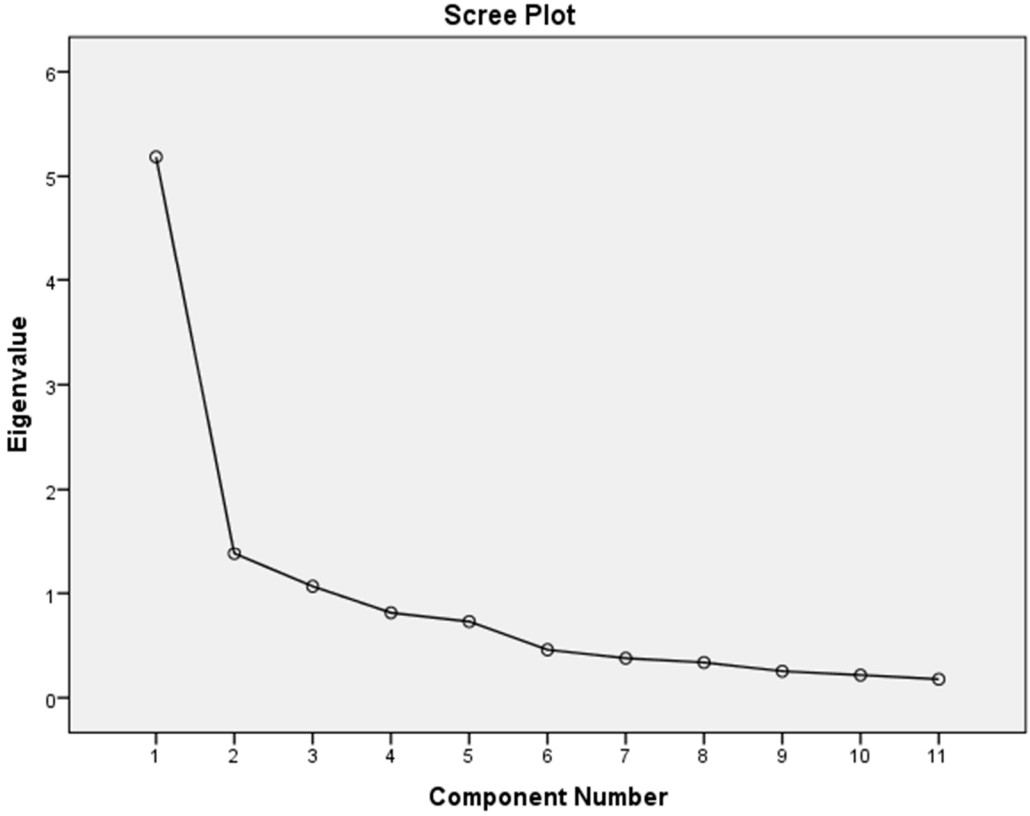

**Figure 1.** Scree plot of a factor structure of sustainable human capital (SHC) methodology.

**Table 1.** SHC methodology factor structure.

|  | Component | |
| --- | --- | --- |
|  | **F1—Uniqueness** | **F2—Positivity** |
| I communicate even if I do not say anything | 0.613 | |
| The meaning of communication is a reaction that it evokes | 0.787 | |
| Every behavior has a positive intention | | 0.738 |
| There is no failure, there is only a feedback | | 0.756 |
| A man works the best way he/she can in a given situation | | 0.887 |
| Mind and body are parts of the same system | 0.810 | |
| We all perceive the world in a unique way | 0.831 | |
| The possibility of choice changes the life | | 0.475 |
| The same method means the same results | 0.700 | |
| When somebody did it, the other can do it as well | | 0.531 |
| I have all resources to be able to make a change | 0.660 | |

| Component | Initial Eigenvalues | | |
| --- | --- | --- | --- |
|  | Total | % of Variance | Cumulative % |
| F1 Uniqueness | 5.186 | 47.142 | 47.142 |
| F2 Positivity | 1.381 | 12.556 | 59.699 |

| Rotation Sums of Squared Loadings | | |
| --- | --- | --- |
| Total | % of Variance | Cumulative % |
| 3.798 | 34.531 | 34.531 |
| 2.768 | 25.168 | 59.699 |

The extracted factors explain 59.699% of variance. This percentage of the explained variance by the extracted factors is acceptable, the factors were able to specify in terms of content.

The inner consistency of individual factors was detected by the calculation of the Cronbach's alpha coefficient (Tables 2 and 3).

**Table 2.** Inner consistency of Uniqueness factor.

| | Scale Mean If Item Deleted | Scale Variance If Item Deleted | Corrected Item-Total Correlation | Cronbach's Alpha If Item Deleted |
|---|---|---|---|---|
| I communicate even if I do not say anything | 20.73 | 16.199 | 0.548 | 0.859 |
| The meaning of communication is a reaction that it evokes | 20.42 | 16.596 | 0.692 | 0.826 |
| Mind and body are parts of the same system | 20.25 | 17.277 | 0.729 | 0.823 |
| We all communicate in a unique way | 20.27 | 16.684 | 0.709 | 0.823 |
| The same method means the same results | 20.73 | 17.660 | 0.617 | 0.840 |
| I have all the resources I need to make a change | 20.54 | 16.018 | 0.654 | 0.833 |

Cronbach's Alpha F1 Uniqueness = 0.858.

**Table 3.** Inner Consistency of Positivity Factor.

| | Scale Mean If Item Deleted | Scale Variance If Item Deleted | Corrected Item-Total Correlation | Cronbach's Alpha If Item Deleted |
|---|---|---|---|---|
| Every behavior has a positive intention | 15.12 | 13.035 | 0.588 | 0.757 |
| There is no failure, there is only a feedback | 15.29 | 11.469 | 0.661 | 0.733 |
| A man works the best he is able in a given situation | 15.50 | 11.592 | 0.669 | 0.730 |
| The possibility of a choice changes a life | 14.63 | 15.654 | 0.478 | 0.792 |
| If somebody did it, the other can do it as well | 14.78 | 13.941 | 0.534 | 0.774 |

Cronbach's Alpha F2 Positivity = 0.798.

The presented Cronbach's Alpha values confirm the satisfactory level of inner consistency of the extracted factors of sustainable human capital development.

Based on the given results, it is possible to regard hypothesis 1 "We assume that it is possible to identify and specify the indicators of sustainable human capital development" as verified.

The connections between the assessment of NLP-C attributes and the indicators of sustainable human capital development.

To analyze the connections between the assessment of NLP-C attributes and the indicators of sustainable human capital development, based on the confirmation of normal data distribution (Tables 4 and 5), Pearson's correlation test was used (Table 6). Data distribution normality was tested with Skewness and Kurtosis.

**Table 4.** Distribution normality of NLP-C factors.

|  | Body Language | Active Listening | Assertiveness | Asking Questions |
|---|---|---|---|---|
| Skewness | −0.688 | −0.559 | −0.893 | −0.574 |
| Kurtosis | 0.356 | −0.506 | 0.646 | −0.225 |

**Table 5.** Distribution normality of SHC factors.

|  | Uniqueness | Positivity |
|---|---|---|
| Skewness | −0.626 | −0.562 |
| Kurtosis | −0.422 | −0.304 |

**Table 6.** Connections between the assessment of neurolinguistic programming-communication (NLP-C) attributes and the indicators of sustainable human capital development using SHC methodology.

|  | Uniqueness | Positivity |
|---|---|---|
| Body language | 0.456 | 0.331 |
| Significance | 0.000 | 0.001 |
| Active listening | 0.572 | 0.333 |
| Significance | 0.000 | 0.001 |
| Assertiveness | 0.437 | 0.452 |
| Significance | 0.000 | 0.000 |
| Asking questions | 0.581 | 0.379 |
| Significance | 0.000 | 0.000 |

The results of the correlation analysis confirmed several statistically significant correlation coefficients. Statistically significant connections were found between all NLP-C attributes and both indicators of sustainable human capital development. All correlation coefficients were positive. This means that the higher the respondents scored in one indicator, the higher they scored in the other one. The respondents who have developed communication skills, as for example the art of asking questions in the right way, who are willing to listen, are able to identify and interpret the nonverbal communication expressions, to express emotions in the authentic way, the communication of their own attitudes and requirements is adequate, at the same time, their body and mind are in harmony and perceive the world in a unique way, they are not afraid of changes since they believe that they have all the needed resources, their direction and intention is positive and perceived in terms of success.

Based on the given results, hypothesis 2 "We assume that there are statistically significant connections between the assessment of selected NLP-C attributes and the indicators of sustainable human capital development" can be considered verified.

The connections between the assessment of NLP-T attributes and the indicators of sustainable human capital development:

To analyze the connections between the assessment of NLP-T attributes and the indicators of sustainable human capital development, based on the confirmation of normal data distribution (Table 7), Pearson's correlation test was used (Table 8).

**Table 7.** Distribution normality of neurolinguistic programming-technique (NLP-T) factors.

|  | Representational Systems | Rapport | Leading |
|---|---|---|---|
| Skewness | −0.081 | −0.781 | −0.064 |
| Kurtosis | −0.229 | −0.517 | −0.882 |

**Table 8.** Connections between the assessment of NLP-T attributes and the indicators of sustainable human capital development using SHC methodology.

|  | Uniqueness | Positivity |
|---|---|---|
| Representational systems | 0.551 | 0.585 |
| Significance | 0.000 | 0.000 |
| Rapport | 0.457 | 0.382 |
| Significance | 0.000 | 0.000 |
| Leading | 0.274 | 0.347 |
| Significance | 0.005 | 0.000 |

The results of the correlation analysis confirmed several statistically significant correlation coefficients. Statistically significant connections were found between NLP-attributes and both indicators of sustainable human capital development. All correlation coefficients were positive. This means that the higher the respondents scored in one indicator, the higher they scored in the other one. The respondents who have their own preferred sensual system, whose communication is one of the principles of success based on co-operation, trust and respect, can adapt themselves and create relationships, at the same time they also communicate effectively without words, they have their body and mind in harmony, perceive the world in a unique way, they are not afraid of changes since they believe that they dispose of all the needed resources, they have a positive direction and the intention that they perceive in terms of success.

Based on the given results, hypothesis 3 "We assume that there are statistically significant connections between the assessment of selected NLP-T attributes and the indicators of sustainable human capital development" can be considered verified.

## 5. Discussion and Conclusions

The main aim of the research was to identify and specify the indicators of sustainable human capital development and their relationship to NLP.

The first indicator of sustainable human capital development showed the feature of uniqueness that can be developed also at a subconscious level. In this context, metaprograms help to understand who has what values and processes information making it easier to achieve success in any area. These metaprograms can be revealed and identified by NLP [24]. It is not appropriate to typecast and evaluate people according to models that complement each other [25]. Like values, the metaprograms are changing, too, depending on a context. Effective communication is not only about words we use [26]. Research shows that we communicate by expressions, body language more than by words we use.

By the second indicator, the feature of positivity was identified. The most important elements within the proper formulation of life goals include their positive formulation [27]. If our verbal and non-verbal behavior supports our goal, we are congruent. The inner congruence gives individuals strength and personal power. It is a tool saying whether we are going in a good, positive direction [28].

Research confirmed that NLP has a positive effect on individuals who completed an NLP course within an experimental course of personality development compared to a control group who did not attend any NLP course. These findings support the sustainability of NLP results in terms of a sustainable life quality [29]. The most research on NLP method was conducted in the context of communication research the development of which has generally a positive influence on the attributes of human capital [30].

The aim of the research was also analyzing the connections between the selected NLP attributes—NLP-communication (NLP-C), NLP-techniques (NLP-T), and the indicators of sustainable human capital development. The results of correlation analysis confirmed statistically significant correlation coefficients. The respondents who developed communication skills as the art of asking questions in a good and proper manner, who are willing to listen, they can identify and

interpret non-verbal communication expressions, to express emotions authentically express emotions, communication of their own attitudes and requirements is adequate, at the same time, from the point of view of sustainable human capital development have body and mind in harmony and perceive the world in a unique way, they dispose of necessary resources, their directions and intentions are positive. The respondents who have their own preferred sensual system, whose communication is one of the principles of success based on co-operation, trust and respect, they can adapt themselves and create relationships, at the same time they also communicate effectively without words, they are not afraid of changes since they believe that they have all the needed resources, they have a positive direction and the intention they perceive in terms of success.

The given findings support the use of the NLP concept from the point of view of sustainable human capital development. NLP is a complex set of models and techniques enabling effective thinking and behavior. This approach has a long-term effect, it is not a one-time but continual process [23]. We can expect the positive effect of NLP in various areas of organizations such as human resources management [31] but also outside organizations in relation to the environment, sustainable regional development [32], fair economic competition, corporate social responsibility [33], as well as other areas in relation to which this issue is discussed, e.g., intellectual capital [34]. An overview of approaches to human capital sustainability in the context of related areas is presented by several authors [35,36].

In this context, NLP is little scientifically researched [37]. It points out the complexity of implementation and in particular, the measurement of NLP results. The fact remains that NLP is used worldwide, and its popularity is increasing. The presented findings point to the possibilities of NLP application in the context of the sustainable development of human capital. The limits of the presented analytical results are determined by several factors. Within the holistic concept of sustainability approach in terms of human capital sustainability, it is necessary to consider a holistic view of the issue and not only summarize the results from individual areas of knowledge. It is necessary to interpret this issue also in the context of economic sustainability [38]. These requirements also relate to the acceptance of the specifics of the cultural patterns of society, the norms and values that people in the given society follow. It is necessary to interpret and generalize the results of the presented research with a high level of caution which is also related to the size and selection of the research sample. The size of the research sample in the presented research project was limited by the fact that only NLP course graduates were included in the sample. At the same time, it should be noted that the use of NLP in the context of sustainable human capital development will need to be experimentally verified in the project with an experimental and control group in the context of ante and post measurement.

In future research projects, attention should be paid to the issues whether the findings are generally, i.e., universally valid, or whether these findings are situationally conditioned. In this context, it is inevitable to emphasize the need of the research of personality specifics from the point of view of NLP application related to human capital sustainability. At the same time, it will be necessary to orientate the research to the analysis of the relations between human capital sustainability on one hand and natural, financial, and economic capital sustainability on the other hand [35].

**Author Contributions:** Investigation, supervision, and resources, M.F. and Z.B.; funding acquisition and project administration, R.Š.; conceptualization, Z.B. and M.F.; formal analysis and software, M.F.; methodology, M.F. and Z.B.; validation, Z.B.; data curation and visualization, M.F. and Z.B.; writing—original draft, review & editing, E.B.

**Funding:** This research was funded by Vedecká Grantová Agentúra MŠVVaŠ SR a SAV (1/0807/19).

**Conflicts of Interest:** The authors declare no conflict of interest. The funders had no role in the design of the study; in the collection, analyses, or interpretation of data; in the writing of the manuscript, or in the decision to publish the results.

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
