# Peer review of "Implementing the Concept of Neurolinguistic Programming Related to Sustainable Human Capital Development"

_sustainability, doi:10.3390/su11154031_

Reviewer 1 Report

The article submitted for review concerns the soft competences of personal development. In the context of numerous concepts of human capital, which common denominator is the understanding of human capital as a factor undoubtedly affecting the efficiency of work, the use of the term human capital development in exchange for the concept of personal development can be considered legitimate. The use of the term “sustainable” human capital development requires more theoretical justification.

After analyzing the course and results of the research, two questions arise. First of all, whether the use of NLP will not adversely affect other features of human capital. Secondly, what is the duration of the positive impact of NLP on human behavior. I would suggest to provide in the last part of the article the possibility of further research on the effectiveness of the use of NLP in organizations that will allow answering these questions. The use of the term "human capital", which is an organizational, economic and social resource, somehow obliges to do so.

Despite the aforementioned remarks, I assess the research positively, as a milestone to the new method of human capital development in organizations.

In addition, in line 140, I noticed an error:

"Representational systems – believing that every human being prefers o (one???) of the five sensory systems".

Author Response

Dear reviewer,

thank you very much for your review and incentive suggestions.

Point 1: The use of the term “sustainable” human capital development requires more theoretical justification.

Response 1: An overview of approaches to human capital sustainability in the context of related areas is presented by several authors [35, 36]. 

Point 2: First of all, whether the use of NLP will not adversely affect other features of human capital.

      Response 2: The most research on NLP method was conducted in the context of communication research the development of which has generally a positive influence on the attributes of human capital [30].

Point 3: What is the duration of the positive impact of NLP on human behavior

Response 3: .NLP is a complex set of models and techniques enabling effective thinking and behaviour. This approach has a long-term effect, it is not a one-time but continual process [23].

Point 4: I would suggest to provide in the last part of the article the possibility of further research on the effectiveness of the use of NLP in organizations that will allow answering these questions.

      Response 4: We can expect the positive effect of NLP in various areas of organizations such as human resources management [31] but also outside organizations in relation to the environment, sustainable regional development [32] fair economic competition, corporate social responsibility [33] as well as other areas in relation to which this issue is discussed, e.g. intellectual capital [34]. An overview of approaches to human capital sustainability in the context of related areas is presented by several authors [35, 36].  In the future research projects, the attention should be paid to the issues whether the findings are generally, i.e. universally valid, or whether these findings are situationally conditioned. In this context, it is inevitable to emphasize the need of the research of personality specifics from the point of view of NLP application related to human capital sustainability. At the same time, it will be necessary to orientate the research to the analysis of the relations between human capital sustainability on one hand and natural, financial, and economic capital sustainability on the other hand [35]. 

    Point 5: In addition, in line 140, I noticed an error: "Representational systems – believing that every human being prefers o (one???) of the five sensory systems".

    Response 5: Representational systems – believing that every human being prefers one of the five sensory systems.

Reviewer 2 Report

The questions  is original and well defined. The results are interpreted appropriately and are significant. The article is written in an appropriate way, the data and analyses are presented appropriately. The analyses are performed with the highest technical standards. The paper will be of interest only to a limited number of people. The work provide an advance towards the current knowledge. The English language appropriate and understandable. Authors should give suggestions for further research in this area.

Author Response

Dear reviewer,

thank you very much for the review and incentive suggestion.

Point 1: Suggestions for the future research in this area:

Response 1: In the future research projects, the attention should be paid to the issues whether the findings are generally, i.e. universally valid, or whether these findings are situationally conditioned. In this context, it is inevitable to emphasize the need of the research of personality specifics from the point of view of NLP application related to human capital sustainability. At the same time, it will be necessary to orientate the research to the analysis of the relations between human capital sustainability on one hand and natural, financial, and economic capital sustainability on the other hand [35].